# Feasibility and Effects of Implementing Multimodal Prehabilitation Before Cytoreductive Surgery in Patients with Ovarian Cancer: The Gynofit Multicenter Study [note 1]

**DOI:** 10.3390/cancers17091393

**Published:** 2025-04-22

**Authors:** Stella van der Graaff, Tessa A. M. Backhuijs, Frank P. de Kort, Elize W. Lockhorst, Huberdina P. M. Smedts, Jennifer M. J. Schreinemakers, Gatske M. Nieuwenhuyzen-de Boer, Janneke S. Hoogstad-van Evert

**Affiliations:** 1Department of Gynecology and Obstetrics, Amphia Hospital, 4818 CK Breda, The Netherlands; svandergraaff@amphia.nl (S.v.d.G.); dsmedts@amphia.nl (H.P.M.S.); 2Department of Sports Medicine, Amphia Hospital, 4818 CK Breda, The Netherlands; tbackhuijs@amphia.nl; 3Department of Physiotherapy, Albert Schweitzer Hospital, 3318 AT Dordrecht, The Netherlands; f.p.dekort@asz.nl; 4Department of Surgery, Amphia Hospital, 4818 CK Breda, The Netherlands; elockhorst@amphia.nl (E.W.L.);; 5Department of Gynecology and Obstetrics, Albert Schweitzer Hospital, 3318 AT Dordrecht, The Netherlands; 6Department of Gynecological Oncology, Erasmus Medical Centre, 3015 GD Rotterdam, The Netherlands

**Keywords:** prehabilitation, feasibility, ovarian cancer, cytoreductive surgery, perioperative care

## Abstract

Multimodal prehabilitation is an emerging approach to optimize patients’ conditions before surgery but has not yet been studied extensively in patients with ovarian cancer. This study evaluated the feasibility and effects of two different multimodal prehabilitation programs implemented before cytoreductive surgery in ovarian cancer patients in two Dutch hospitals. We observed high eligibility and participation rates, along with improved functional capacity following prehabilitation. These findings suggest that prehabilitation is both feasible and effective for patients with ovarian cancer, even during neoadjuvant chemotherapy.

## 1. Introduction

In 2022, 325,000 women worldwide were diagnosed with ovarian cancer (OC), representing 23% of all gynecological cancers and 3.4% of new cancer cases in women globally [1,2]. Because the non-specific symptoms of OC often lead to a delayed diagnosis, 60% of the women present with advanced-stage disease [3]. Cytoreductive surgery (CRS) is a cornerstone of advanced-stage OC treatment and is often extensive, including hysterectomy, omentectomy and in 25% bowel surgery. Many patients may have risk factors such as high age, high body mass index (BMI), high American Society of Anesthesiologists (ASA) score, comorbidities and malnutrition, which make them more susceptible to intra- and postoperative complications [4,5,6]. Major complications occur in up to 46% of the patients, depending on the surgical complexity, the patient’s condition and whether the surgery is primary cytoreductive or secondary to neoadjuvant chemotherapy [5,7,8]. These complications can prolong hospital stays, delay the initiation of adjuvant chemotherapy—negatively impacting progression-free and overall survival—and contribute to increased morbidity, potential mortality and reduced quality of life [9,10,11,12].

Prehabilitation embraces the concept of “better in, better out” by optimizing a patient’s condition before surgery to enhance recovery and treatment outcomes. It has been shown to improve chemotherapy tolerance in esophageal and gastric cancer [13,14]. Improvement of chemotherapy tolerance is highly needed in OC, as delays or dose reduction occurs in 31–48% of the patients and is associated with decreased progression-free and overall survival [15,16,17]. Physical activity, quitting smoking before and after an OC diagnosis and a high-quality diet are all linked to improved survival [18,19,20]. Multimodal prehabilitation, which combines physical exercise, dietary optimization, and quitting smoking, targets these modifiable behaviors that are linked to a better prognosis and may improve long-term outcomes.

Multimodal prehabilitation is an emerging concept and has been demonstrated to be beneficial in several surgical fields. The primary evidence on prehabilitation prior to abdominal surgery comes from gastrointestinal surgery studies [21,22,23,24,25]. For example, in the PREHAB study, a 4-week standardized supervised multimodal prehabilitation program significantly reduced the number and severity of complications in 251 patients undergoing gastrointestinal surgery for low-stage colorectal cancer [25]. These results are promising and might also apply to cytoreductive surgery in ovarian cancer. However, there is yet little evidence of multimodal prehabilitation in gynecologic oncology.

There are only three primary studies published evaluating the postoperative effects of multimodal prehabilitation in the gynecologic oncology population [26,27,28]. Two studies implemented a multimodal prehabilitation program for patients with advanced stage OC prior to cytoreductive surgery [26,27] and another for patients with FIGO stage IA to IIIC endometrial cancer undergoing laparoscopy [28]. These studies found a significantly shorter hospital stay, a shorter time to start adjuvant chemotherapy, fewer readmissions, an earlier return to a normal diet and a lower need for intraoperative blood transfusions [26,27,28]. Two studies have reported on the feasibility and safety of the implementation of multimodal prehabilitation in gynecological oncology patients. Dhanis et al. included 111 patients with ovarian, endometrial and vulvar cancer in an academic hospital with high recruitment rates; 40 of them were patients with OC [29]. Diaz-Feijoo et al. included 20 patients with OC and found an eligibility rate of 75% [26]. Both studies did not report any major adverse events [26,29], suggesting that prehabilitation is safe in OC patients.

Although these results are promising, the evidence on prehabilitation in OC patients is limited and only includes studies with small numbers of patients. Therefore, the aim of this study is to evaluate the real-life feasibility and effects of implementing a multimodal prehabilitation program before cytoreductive surgery in OC patients in two top clinical hospitals in the Netherlands.

This paper is an extended version of our abstract *Multimodal Prehabilitation in Ovarian Cancer Patients Results in Improved Chemotherapy Tolerance: A Multicenter Cohort Study* [30]. This abstract was published as E-poster for the ESGO 2025 Congress, Rome, Italy, 20–23 February 2025.

## 2. Materials and Methods

### 2.1. Patient Recruitment

This multicenter pilot study included patients from two top Dutch clinical hospitals. The Amphia Hospital in Breda enrolled all OC patients who underwent cytoreductive surgery between April 2023 and April 2024. The Albert Schweitzer Hospital in Dordrecht enrolled all OC patients who underwent neoadjuvant chemotherapy before cytoreductive surgery between January 2023 and March 2024.

In both hospitals, the patients who participated in the prehabilitation program were included in the prehabilitation group, and the patients who did not participate in the program during the study period were included in the non-prehabilitation group. All OC patients who underwent cytoreductive surgery at the Amphia Hospital in 2022 were included in a control group and served as the control group.

### 2.2. Multimodal Prehabilitation Programs

Both hospitals offered a multimodal prehabilitation program that varied in timing and extent but generally overlapped in content (Figure 1).

At the Amphia Hospital, the prehabilitation program started a few weeks before cytoreductive surgery and lasted 3 weeks. The program started with an intake by a doctor’s assistant at a dedicated prehabilitation outpatient clinic and was based on four pillars:Physiotherapy: all patients were referred to an outpatient physiotherapist close to their home for individually supervised low-intensity strength and cardio training sessions three times a week. This is in accordance with the Fit4Surgery protocol [31], which was also described in the prehabilitation study by Dhanis et al. [25,29]. Baseline and end-line measurements of the physiotherapy trajectory included the Maximum Short Exercise Capacity (MSEC) on the modified Steep Ramp Test (mSRT) and the One Repetition Maximum (1RM) on the leg press.Dietary and lifestyle advice: all patients received verbal and written advice on a healthy diet (e.g., increased protein intake) and lifestyle. Patients were referred to a dietician if deemed necessary based on screening with the Patient-Generated Subjective Global Assessment (PG-SGA) and Short Nutritional Assessment Questionnaire (SNAQ) score.Mental well-being and sleep: all patients received verbal and written advice on mental well-being and sleep hygiene. Patients were advised to use a free app on their own mobile device to access yoga-, mindfulness- and meditation exercises at home. Patients with psychological symptoms who wanted support were referred to a social worker or medical psychologist.Intoxication: active smokers were referred to a smoking cessation coach. Excessive alcohol or drug users were referred for first-line addiction treatment after consulting their general practitioner.In addition, patients were referred to a geriatrician or sports medicine specialist when needed, based on screening with the estimated VO2 max, Geriatric 8 (G8) screening tool, grip strength, Timed Up and Go Test (TUG), Timed Chair Stand Test and Six-Item Cognitive Impairment Test (6CIT).

At the Albert Schweitzer Hospital, the prehabilitation program started at the same time as neoadjuvant chemotherapy and lasted nine weeks. This program is based on the PADOVA trial, where OC patients undergoing (neo)adjuvant chemotherapy were randomized between exercise and dietary intervention or standard care [32,33].

Our prehabilitation program was based on four pillars:Physiotherapy: all patients were referred to the physiotherapist for individual supervised training sessions. They were first seen by an in-hospital physiotherapist who conducted baseline measurements, including the 6-Minute Walk Test (6MWT), the modified Steep Ramp Test (mSRT) and a 1RM on the leg press, low row, chest press and lateral pulldown. After an electrocardiogram (ECG) was administered, patients were referred to a first-line physiotherapist near the hospital for supervised training following a structured protocol: one low-intensity strength and cardio training 5 days after receiving chemotherapy and two high-intensity strength and cardio training per week during the second and third week of the chemotherapy cycle. Since patients received 3 cycles of neoadjuvant chemotherapy, this protocol was repeated twice. A few days before the cytoreductive surgery, patients returned to the in-hospital physiotherapist for endpoint measurements.Dietary advice: all patients were referred to an in-hospital dietician. The dietician performed baseline measurements of the Patient-Generated Subjective Global Assessment (PG-SGA), grip strength, weight, body mass index (BMI), muscle mass and fat mass at the intake and performed endpoint measurements a few days before cytoreductive surgery.Quitting smoking: active smokers were referred to a smoking cessation coach.Mental health: if deemed necessary based on our screening, patients were referred to a social worker or medical psychologist.

At both hospitals, patients were advised to do low-intensity activities such as walking or biking for at least one hour a day on the days they did not attend physiotherapy.

### 2.3. Baseline, Clinical and Surgical Characteristics

Baseline, clinical and surgical characteristics were collected from all patients of the prehabilitation, control and non-prehabilitation groups. Comorbidities were scored with the Age-Adjusted Charlson Comorbidity Index (CCI) [34,35]. This index predicts the 10-year survival of patients based on their age and comorbidities, such as (metastasized) solid tumors [34,35] and has also been validated for OC patients undergoing cytoreductive surgery [36]. The surgical complexity is scored with the surgical complexity scoring system of Aletti et al., which is developed specifically for patients with advanced OC [37].

### 2.4. Study Outcomes

#### 2.4.1. Primary Outcomes

The primary objective of this study was to evaluate the feasibility of implementing a multimodal prehabilitation program in OC patients before cytoreductive surgery. Primary outcomes are eligibility rates, participation rates and adherence to the physiotherapy sessions. Adherence to the physiotherapy sessions was considered satisfactory if a patient attended ≥75% of the targeted physiotherapy sessions.

#### 2.4.2. Secondary Outcomes

The secondary objective of this study was to evaluate the effects of the multimodal prehabilitation program, assessed across three domains:Functional capacity: the baseline and endpoint measurements of the physiotherapy trajectory within the prehabilitation groups were compared. Baseline and endpoint measurements of the physiotherapy program included the MSEC on the modified Steep Ramp Test and the 1RM on the leg press for both hospitals. For the Albert Schweitzer Hospital, the baseline and endpoint measurements also included the 1RM on the low row, chest press and lateral pulldown and the walking distance with the 6-Minute Walk Test (6MWT).Postoperative outcomes: The postoperative outcomes of the Amphia prehabilitation and control groups were compared. These included the length of stay of the postoperative hospital and intensive care unit, the 90-day complications, readmission and reoperation rates, discharge destination, and time to initiate adjuvant chemotherapy. Complications were scored with the Comprehensive Complication Index (CCI) score [38,39], based on the Clavien-Dindo classification [40].Adjuvant chemotherapy dose modifications: the rates of adjuvant chemotherapy dose reduction and deferrals between the Amphia prehabilitation and control groups were compared. Dose reduction was defined as a paclitaxel reduction of ≥15% or a carboplatin area under the curve (AUC) reduction of ≥1. Dose deferral is defined as a delay of ≥7 days within the adjuvant chemotherapy cycles.

### 2.5. Statistical Analysis

As this study evaluates the efficacy of prehabilitation, a per-protocol analysis was used. Data was analyzed with IBM SPSS Statistics, version 29.0.2.0. Continuous outcomes were tested for normality and presented as mean and standard deviation or as median and interquartile range, depending on normality. Continuous outcomes were compared with the unpaired *t*-test or Mann-Whitney U test, respectively. Categorical variables were presented as frequencies and proportions and were compared with the Chi-squared test or Fisher’s exact test. Baseline and endpoint measurements of the physiotherapy trajectory were compared with the paired *t*-test after being tested for normality. The significance level was set at *p* = 0.05. No sample size calculation or a priori power analysis was made as the primary objective of this study was to evaluate feasibility. A post-hoc power analysis was performed using G*Power, version 3.1, to analyze the statistical power of the observed postoperative differences, given the sample sizes. The effect size was set at 0.5, the α level at 0.05 and the minimally required power was set at 80%. Missing data were excluded.

## 3. Results

### 3.1. Feasibility

At the Amphia Hospital, 28 patients with OC were initially scheduled for cytoreductive surgery during the study period (Figure 2). Three patients were ineligible for prehabilitation; two had surgery scheduled within three weeks, making prehabilitation unfeasible, and one declined participation due to the cost of physiotherapy treatments. This results in an eligibility rate of 89%. Six out of the 25 (24%) eligible patients did not participate in the prehabilitation program; four were not referred by their gynecologist for unspecified reasons, and two preferred to undergo self-directed physiotherapy. These patients, along with the ineligible patients, were assigned to the non-prehabilitation group. With 19 out of 28 patients participating in the prehabilitation program, the participation rate was 68%. Following intake at the prehabilitation outpatient clinic, 15 of 19 patients (79%) were referred to the physiotherapist. Of the four who were not referred, two patients had surgery within three weeks, one was already enrolled in a physiotherapy program, and one patient was considered too fit for physiotherapy. Among the 15 referred patients, three did not attend any training sessions: one delayed participation until the surgery was confirmed but had less than three weeks remaining, one was hospitalized due to weakness after neoadjuvant chemotherapy, and one did not schedule physiotherapy appointments for unknown reasons. One patient in the prehabilitation group did not undergo cytoreductive surgery due to disease progression during chemotherapy but remained included in the prehabilitation group, although no appointment for the endpoint measurement was scheduled. In the end, 11 prehabilitation patients underwent both surgery and the full prehabilitation program.

At the Albert Schweitzer Hospital, the eligibility rate was 83%. Of the 18 patients scheduled for neoadjuvant chemotherapy and cytoreductive surgery, three patients were ineligible: two due to frailty and one due to unwillingness to participate (Figure 3). Two eligible patients did not participate in the prehabilitation program; one patient was not referred by the gynecologist for unknown reasons, and the other dropped out after the intake due to transportation preferences. As 13 of the 18 eligible patients participated in the prehabilitation program, the participation rate was 72%. The two eligible patients who did not undergo prehabilitation, along with the ineligible patients, were assigned to the non-prehabilitation group. Ultimately, 13 patients followed the program, but five did not complete the trajectory. Two patients did not undergo surgery due to extensive disease found during diagnostic laparoscopy, one died after the first cycle of neoadjuvant chemotherapy (severely obese (BMI 47), likely due to massive pulmonary embolisms), one missed the endpoint measurement appointment and one discontinued early. The latter had Steinert myotonic dystrophy with chronic hypercapnia. Physiotherapy was too exhausting for her, and she was considered too high-risk for surgery.

### 3.2. Study Population

The Amphia cohort consisted of a prehabilitation group of 19 patients, a control group of 15 patients and a non-prehabilitation group of nine patients. Their baseline, clinical and surgical characteristics are presented in Appendix A. No significant differences were observed between the three groups.

The Albert Schweitzer cohort consisted of a prehabilitation group of 13 patients and a non-prehabilitation group of five patients. Aside from a significant difference in tumor histology type, no other significant differences were found between the prehabilitation and non-prehabilitation groups (see Appendix A).

### 3.3. Physiotherapy Program

#### 3.3.1. Adherence

At the Amphia Hospital, 11 prehabilitation patients underwent surgery and the full physiotherapy program. Their mean number of physiotherapy sessions was 7.9 (SD 3.8, range 2 to 14). Six out of 11 patients (55%) completed ≥75% of the target number of nine training sessions. Three patients exceeded the target of nine training sessions as they began physiotherapy more than three weeks before surgery, allowing for program extension.

At the Albert Schweitzer Hospital, of the eight patients who had baseline and endpoint measurements available, the mean number of training sessions was 15.0 (SD 7.7, range 4 to 27). Five out of eight patients (63%) completed ≥75% of the target number of 15 training sessions. Due to delayed neoadjuvant chemotherapy cycles, four patients exceeded the target of 15 training sessions, extending their prehabilitation beyond nine weeks.

#### 3.3.2. Effects

At the Amphia Hospital, 11 prehabilitation patients underwent the full physiotherapy program and surgery. However, only six patients had both baseline and endpoint measurements available; three patients dropped out because they found the program too intensive, one patient had surgery earlier than planned, and one patient had surgery later than planned and was lost to follow-up for endpoint measurements. The MSEC on the mSRT was significantly higher at the endpoint compared to the baseline (mean difference 31.7, 95% CI 5.6 to 57.7, *p* = 0.03, Table 1). The 1RM for the leg press was not significantly higher at the endpoint compared to baseline (mean difference 18.5 kg, 95% CI −8.9 to 45.9, *p* = 0.14).

At the Albert Schweitzer Hospital, eight patients underwent the full physiotherapy program and surgery. The MSEC on the mSRT was significantly higher at the endpoint compared to baseline (mean difference 17.2, 95% CI 3.1 to 31.2, *p* = 0.03), while the 1RM on the leg press was not (mean difference 12.8, 95% CI −0.2 to 25.7, *p* = 0.053). The 1RM for the low row (mean difference 4.9, 95% CI 2.6 to 7.2, *p* = 0.002), chest press (mean difference 6.0, 95% CI 1.5 to 10.5, *p* = 0.02) and lateral pulldown (mean difference 4.4, 95% CI 3.0 to 5.9, *p* < 0.001) were all significantly higher at the endpoint measurement. The 6MWT distance showed a mean difference of 39.1 m (95% CI −3.7 to 61.8, *p* = 0.07), which was not significant.

Combining measurements from both hospitals demonstrated a significant increase in the mean endpoint measurements of the MSEC and 1RM for the leg press compared to the baseline.

### 3.4. Postoperative Outcomes

At the Amphia Hospital, 18 prehabilitation patients and 15 control patients underwent cytoreductive surgery. The hospital stay, ICU stay, number and severity of complications, readmission and reoperation rates and discharge destination did not significantly differ between the two groups.

Of the prehabilitation group, 14 out of 18 patients received adjuvant chemotherapy (78%). Of the control group, 13 out of 15 patients received adjuvant chemotherapy (87%). Dose reduction due to side effects was required in three out of 14 patients (21%) in the prehabilitation group (two for polyneuropathy, one for general malaise) and in eight out of 13 patients (73%) in the control group (five for polyneuropathy, three for hematologic toxicity), showing a significant difference (*p* = 0.017). However, four control group patients received six cycles of adjuvant chemotherapy without prior neoadjuvant chemotherapy, while all prehabilitation patients had a maximum of three cycles. When analyzing dose modifications within the first three adjuvant cycles, the difference was no longer significant (21% vs. 55%, *p* = 0.12). Dose referral rates did not significantly differ between groups.

A post-hoc power analysis of the postoperative outcomes with 18 prehabilitation and 15 control patients shows a power of 39%. The power of differences in chemotherapy dose modifications in 14 prehabilitation and 13 control patients is 34%.

At the Albert Schweitzer Hospital, seven prehabilitation patients underwent cytoreductive surgery. Their postoperative outcomes can be seen in Table 2.

## 4. Discussion

This multicenter study evaluated the feasibility and effects of implementing multimodal prehabilitation before cytoreductive surgery in OC patients. Moreover, this study is the first to examine adjuvant chemotherapy tolerance after prehabilitation in OC patients. Our findings show that prehabilitation is both feasible and effective for patients with OC undergoing cytoreductive surgery, even during neoadjuvant chemotherapy.

Evidence on the feasibility of implementing multimodal prehabilitation in OC is emerging. Two reports on prehabilitation in OC are published, with comparable results. Dhanis et al. found that in 40 OC patients, the recruitment rate was 73%, and the participation rate was 67% [29]. Adherence to the supervised exercise program was 85%, and adherence to the other parts of the multimodal prehabilitation program varied from 22% for smoking cessation to 100% for psychologist appointments [29]. Diaz-Feijoo et al. included 20 patients with OC and found an eligibility rate of 75% [26]. Adherence to physical training, nutritional optimization and psychological sessions was satisfactory in 87%, 100% and 80% of the patients, respectively [26]. These eligibility and participation rates are comparable to the ones found in this study, with partially overlapping reasons for ineligibility, non-participation or dropout [26,29].

This multicenter study contributes to the available evidence by highlighting the feasibility of implementing prehabilitation in OC patients and by showing the effect on tolerance for adjuvant chemotherapy. Patients demonstrated improved functional capacity after prehabilitation. The Albert Schweitzer prehabilitation group started at a lower functional capacity compared to the Amphia group. This difference is likely because the Albert Schweitzer patients began prehabilitation alongside neoadjuvant chemotherapy when patients were at their weakest. However, both groups showed similar percentage improvements, suggesting that prehabilitation enhances functional capacity regardless of its timing.

This study shows that implementing multimodal prehabilitation is feasible in OC patients before cytoreductive surgery, including candidates for primary surgery. Since the interval between the cancer diagnosis and primary cytoreductive surgery usually spans a few weeks due to adjuvant radiologic imaging, multidisciplinary consultations and preoperative assessments, a prehabilitation program of at least three weeks could be accommodated with minimal, if any, delay to surgery, without significantly reducing eligibility rates nor affecting prognosis.

This study found no statistical differences in postoperative outcomes. This is most likely a result of the small sample size and the low number of events, which consequently led to limited statistical power. As shown in the post-hoc power analysis, the achieved power was 39%, which is well below the minimally required threshold of 80%. This absence of differences in postoperative outcomes is in line with findings from the existing literature. Diaz-Feijoo et al. [26] reported a significantly shorter hospital stay and a reduced time to adjuvant chemotherapy initiation, whereas Miralpeix et al. [27] found no significant differences in postoperative outcomes. Notably, no solid evidence for prehabilitation to cause a reduction in postoperative complications in OC patients is available yet, unlike studies in colorectal cancer. Molenaar et al. reported a significant reduction in both the number and severity of postoperative complications after prehabilitation in a cohort of 251 colorectal cancer patients [25]. However, the PREHAB study included larger patient cohorts, increasing statistical power to detect significant effects [25].

A key strength of this study is the assessment of adjuvant chemotherapy tolerance after prehabilitation. The findings indicate that prehabilitation patients required significantly fewer dose reductions compared to control patients (73% vs. 21%, *p* = 0.017). A possible explanation could be improved cardiopulmonary fitness after prehabilitation, as shown in this article and in the literature [13,14,41], which results in better resilience. This study is the first to report on chemotherapy tolerance in OC, as the effects of prehabilitation on chemotherapy tolerance have previously only been studied in esophagogastric and breast cancer patients [13,14,41]. Due to the multicenter aspect, this study gives a detailed report on the feasibility and effects of implementing multimodal prehabilitation in OC patients in top clinical oncology hospitals in the Netherlands, thereby providing valuable insights into the practical, real-life aspects of implementing prehabilitation.

A limitation of this study is the small sample size and omission of control patients from the Albert Schweitzer Hospital, despite including nearly all newly diagnosed OC patients from the two hospitals. Larger multicenter trials are needed to investigate the effects of prehabilitation in OC patients further. Another limitation is the relatively low adherence to the physiotherapy program, which may have impacted the overall effectiveness of prehabilitation. However, this study still showed improved functional capacity after prehabilitation, suggesting that prehabilitation can positively influence patients’ physical capacity even with suboptimal adherence. Besides that, only a selected number of patients had complete physiotherapy data, which could have introduced bias and decreased statistical power. At last, this study did not focus on quality of life or long-term outcomes such as progression-free survival. These aspects should be investigated in future research.

Multimodal prehabilitation is a valuable opportunity to introduce OC patients to a healthier lifestyle. A healthy lifestyle, including physical exercise, a high-quality diet, and not smoking, before and after diagnosis is associated with improved survival outcomes in patients with OC [18,19,20]. However, studies on lifestyle interventions after an OC diagnosis remain limited, particularly those evaluating combined lifestyle interventions. The results of a large trial investigating the survival benefits of a diet and physical activity intervention post-diagnosis are awaited [42].

Additionally, future research should include large randomized trials to adequately assess the postoperative benefits of prehabilitation in gynecologic oncology patients, which is currently investigated in three ongoing trials: SOPHIE, PROPER and KORE-INNOVATION [43,44,45]. Given their design and sample size, these trials are expected to be of high methodological quality and to adequately detect differences in postoperative outcomes with enough statistical power, unlike our study. However, our study is unique in describing the feasibility and real-time implementation of prehabilitation and in presenting results about adjuvant chemotherapy tolerance.

Furthermore, research should explore the experiences of patients and healthcare professionals, for example, through semi-structured interviews in focus groups and implementing change with implementation tools like RE-AIM or CFIR to optimize prehabilitation programs based on the needs of all stakeholders. We recommend the development of a standardized protocol for implementing prehabilitation care in OC patients in the Netherlands, similar to the existing protocol for colorectal cancer patients [31], to reduce inter-site variability and ensure consistent quality of care.

## 5. Conclusions

This multicenter pilot study showed that multimodal prehabilitation before cytoreductive surgery in patients with OC is both feasible and effective. Patients demonstrated improved functional capacity after the prehabilitation program and greater tolerance for adjuvant chemotherapy compared to control patients. No differences in postoperative outcomes were found, which was anticipated, given the limited statistical power. This study was the first to examine adjuvant chemotherapy tolerance after prehabilitation in OC patients.

## Figures and Tables

**Figure 1 cancers-17-01393-f001:**
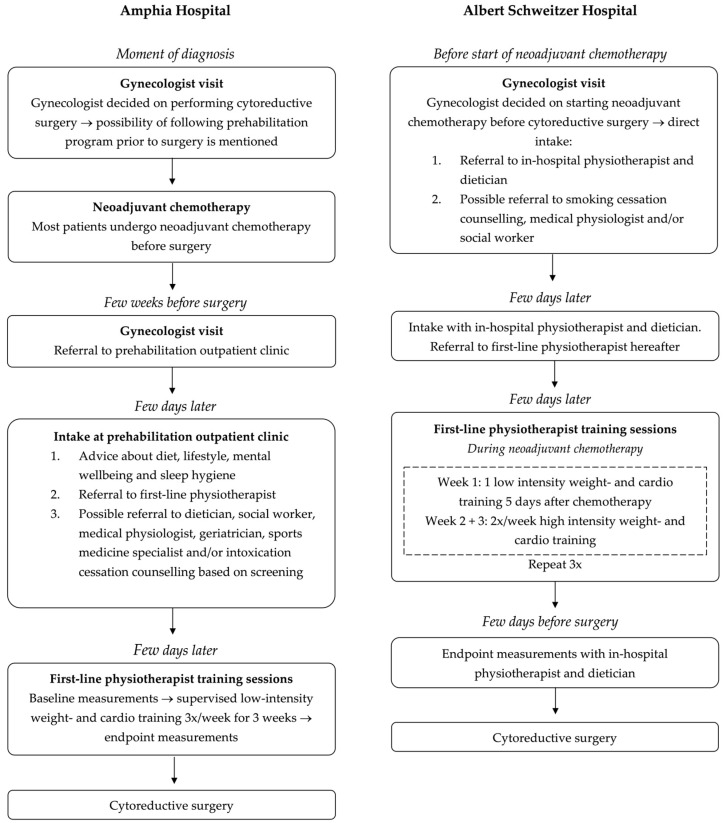
Overview of prehabilitation trajectory in both hospitals.

**Figure 2 cancers-17-01393-f002:**
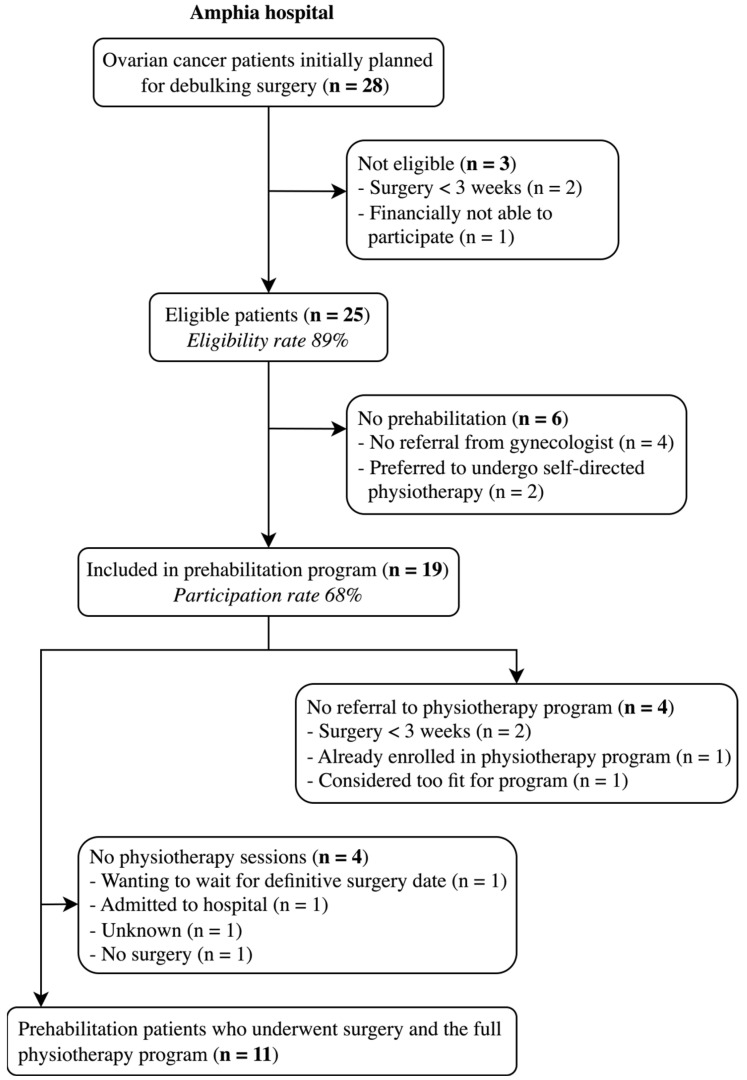
Flow diagram of patient recruitment at the Amphia Hospital.

**Figure 3 cancers-17-01393-f003:**
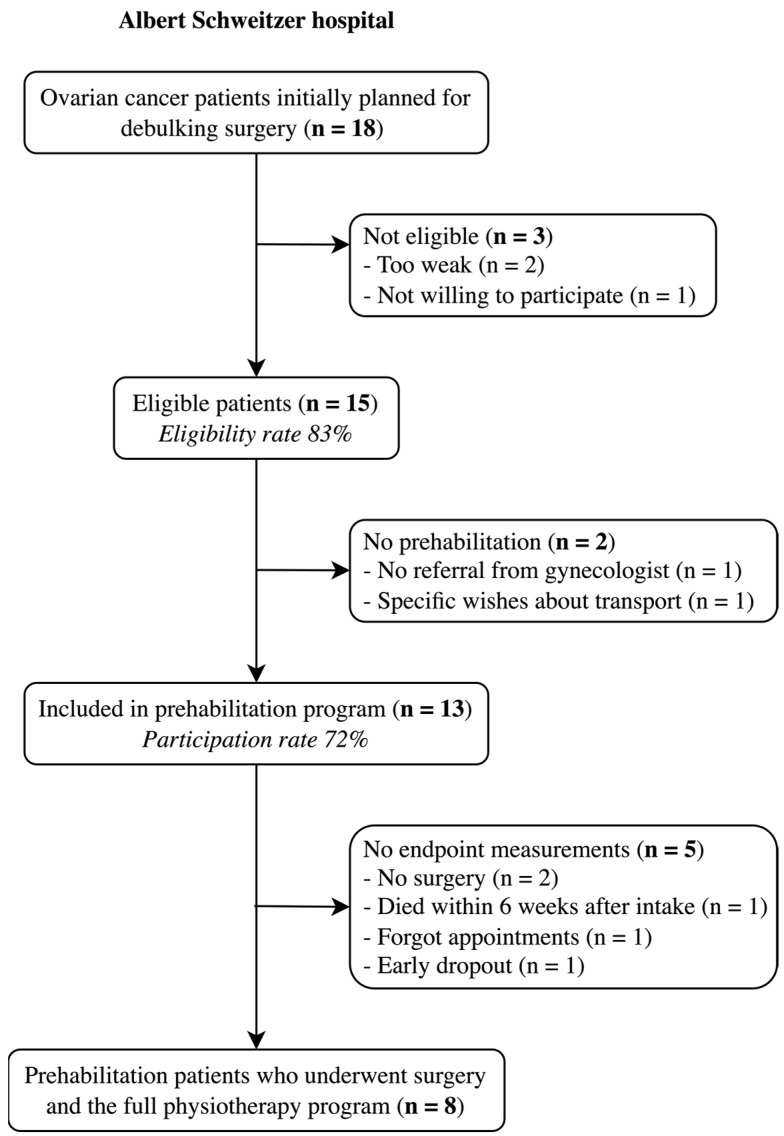
Flow diagram of patient recruitment at the Albert Schweitzer Hospital.

**Table 1 cancers-17-01393-t001:** Baseline and endpoint measurements of the physiotherapy trajectory in both hospitals. Significant findings are in bold, followed by an asterisk. Abbreviations: MSEC = Maximum Short Exercise Capacity, mSRT = modified Steep Ramp Test, 1RM = One Repetition Maximum, 6MWT = 6-Minute Walk Test.

** *Amphia* **	**Patients**	**Baseline** ** *Mean (SD)* **	**Endpoint** ** *Mean (SD)* **	**Mean Difference** ** *Mean (95% CI)* **	**Percentage Change** ** *Mean* **	***p*-Value**
**MSEC mSRT** (watt)	*n* = 6	141.7 (87.6)	173.3 (79.5)	31.7 (5.6 to 57.7)	22.3%	**0.03 ***
**1RM leg press** (kg)	*n* = 6	103.2 (53.3)	121.7 (57.5)	18.5 (−8.9 to 45.9)	17.9%	0.14
** *Albert Schweitzer* **	**Patients**	**Baseline** ** *Mean (SD)* **	**Endpoint** ** *Mean (SD)* **	**Mean Difference** ** *Mean (95% CI)* **	**Percentage Change *Mean***	***p*-Value**
**MSEC mSRT** (watt)	*n* = 6	119.5 (29.7)	136.7 (24.6)	17.2 (3.1 to 31.2)	14.4%	**0.03 ***
**1RM leg press** (kg)	*n* = 8	69.8 (19.9)	82.5 (29.2)	12.8 (−0.2 to 25.7)	18.2%	0.053
**1RM low row** (kg)	*n* = 8	22.4 (4.9)	27.3 (5.1)	4.9 (2.6 to 7.2)	21.9%	**0.002 ***
**1RM chest press** (kg)	*n* = 8	26.8 (7.1)	32.8 (7.9)	6.0 (1.5 to 10.5)	23.1%	**0.02 ***
**1RM lateral pulldown** (kg)	*n* = 8	19.6 (3.0)	24.1 (4.0)	4.4 (3.0 to 5.9)	23.0%	**<0.001 ***
**6MWT walking distance** (meters)	*n* = 8	465.1 (87.7)	494.2 (65.7)	39.1 (−3.7 to 61.8)	6.3%	0.07
**Grip strength** (kg)	*n* = 6					
Right		29.2 (6.0)	25.8 (10.3)	−3.3 (−11.1 to 17.7)	−11.6%	0.58
Left		28.3 (3.9)	27.2 (2.9)	−1.2 (−1.4 to 3.7)	−3.9%	0.29
** *Combined* **	**Patients**	**Baseline** ** *Mean (SD)* **	**Endpoint** ** *Mean (SD)* **	**Mean difference** ** *Mean (95% CI)* **	**Percentage change *Mean***	***p*-Value**
**MSEC mSRT** (watt)	*n* = 12	130.6 (63.4)	155.0 (59.3)	24.4 (11.4 to 37.4)	18.7%	**0.002 ***
**1RM leg press** (kg)	*n* = 14	84.1 (40.0)	99.3 (46.2)	15.1 (3.7 to 26.8)	18.1%	**0.014 ***

**Table 2 cancers-17-01393-t002:** Postoperative outcomes of the Amphia cohort. Significant findings are in bold, followed by an asterisk. Abbreviations: ICU = intensive care unit, CCI = Comprehensive Complication Index.

*Postoperative Outcomes*	Amphia			Albert Schweitzer
	Prehabilitation	Controls	*p*-Value	Prehabilitation *n* = 7
	*n* = 18	*n* = 15		
**Hospital stay** **(days) *median (IQR)***	6.0 (4.5–9.0)	6.0 (4.0–8.0)	0.64	5.0 (3.0–7.0)
Death during a hospital stay	*n* = 1	-		-
**ICU stay after surgery**	*n* = 9 (50%)	*n* = 10 (67%)	0.48	-
Duration of stay (days) *median (IQR)*	1.0 (1.0–3.0)	1.0 (1.0–1.0)	0.34	
**Complications < 90 days**	*n* = 15 (83%)	*n* = 13 (87%)	1.00	*n* = 3 (43%)
Number of complications per patient *median (IQR)*	2.5 (1.0–3.3)	2.0 (1.0–4.0)	0.86	1.0 (1.0–4.5)
Severe complications (Clavien-Dindo ≥ IIIa)	*n* = 5 (28%)	*n* = 5 (33%)	1.00	-
CCI score *median (IQR)*	30.8 (8.7–42.0)	24.2 (20.9–40.2)	0.82	8.7 (0–30.8)
**Readmission < 90 days**	*n* = 3 (17%)	*n* = 3 (20%)	1.00	*n* = 1 (14%)
**Reoperation < 90 days**	*n* = 3 (17%)	*n* = 2 (13%)	1.00	-
**Destination after discharge ^1^**				
Home	*n* = 11 (61%)	*n* = 11 (73%)	1.00	*n* = 5 (71%)
Home with home care	*n* = 5 (28%)	*n* = 4 (27%)		*n* = 2 (29%)
Care hotel	*n* = 1 (6%)	-		-
**Adjuvant chemotherapy**	*n* = 14 (78%)	*n* = 13 (87%)	0.68	*n* = 6 (86%)
Time to start (days) *mean (±SD)*	37.8 (9.2)	42.7 (7.2)	0.15	33.2 (4.7)
Dose modifications ^2^				
Dose reduction (within 3 cycles/within 6 cycles)	*n* = 3 (21%)/-	*n* = 6 (55%)/*n* = 8 (73%)	0.12/**0.017 ***	*n* = 4 (67%) ^3^
Deferral (within 3 cycles/within 6 cycles)	*n* = 5 (36%)/-	*n* = 4 (36%)/*n* = 5 (45%)	1.00/0.70	*n* = 1 (17%) ^3^

^1^ Missing *n* = 1. ^2^ Two patients of the Amphia control group received adjuvant chemotherapy in another hospital and were therefore not included in these rates. ^3^ Both within 3 cycles.

## Data Availability

The data presented in this study are available on request from the corresponding author due to ethical considerations.

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
