# Peer review of "Feasibility and Effects of Implementing Multimodal Prehabilitation Before Cytoreductive Surgery in Patients with Ovarian Cancer: The Gynofit Multicenter Studyâ€"

_cancers, 2025, doi:10.3390/cancers17091393_

Round 1
Reviewer 1 Report
Comments and Suggestions for Authors
FEASIBILITY AND EFFECTS OF IMPLEMENTING MULTIMODAL PREHABILITATION BEFORE CYTOREDUCTIVE SURGERY IN PATIENTS WITH OVARIAN 3 CANCER: THE GYNOFIT MULTICENTER STUDY
Cancers 2025
In this paper, the Authors present data on the feasibility and effects of implementing two different multimodal prehabilitation programs before cytoreductive surgery in patients with ovarian cancer in two Dutch hospitals.
The paper is well written and the English language is appropriate and understandable.
The clinical topics are fascinating because prehabilitation which combines physical exercise, dietary optimization and quitting smoking, is an emerging approach to optimize patients’ conditions before surgery to enhance recovery and treatment outcomes in terms of length of hospital stay, time to start adjuvant chemotherapy, rate of readmissions, return to a normal diet, and a lower need for intraoperative blood transfusion as it has been shown for example in colorectal surgery. To date, only a few data are available in patients with ovarian cancer before cytoreduction surgery. Current published literature includes studies with small numbers of patients. Furthermore, there is yet little evidence on the role of multimodal prehabilitation in improving chemotherapy tolerance in gynecologic oncology.
The Authors observed high eligibility and participation rates, with improved functional capacity following prehabilitation. No significant differences in postoperative outcomes were found between prehabilitation and control patients. Prehabilitation patients appeared to have better tolerance to adjuvant chemotherapy, with fewer dose reductions and dose deferrals than the control group.
The authors thoroughly report on the limitations and bias of this study, including the small sample size, omission of control patients, and relatively low adherence to the full prehabilitation program.
Specific comments:
Do the authors believe that the duration of at least 3 weeks of prehabilitation could reduce the eligibility rates of patients who are candidates for primary cytoreductive surgery?
Could the Authors report the possible reasons for better tolerance to chemotherapy in terms of dose reductions and dose deferrals between prehabilitation patients and control groups?
Author Response
Thank you very much for the nice words on our manuscript and for taking the time to review this manuscript. Please find the detailed responses below and the corresponding revisions, marked in red, in the re-submitted files.
Comments 1: Do the authors believe that the duration of at least 3 weeks of prehabilitation could reduce the eligibility rates of patients who are candidates for primary cytoreductive surgery?
Response 1: Thank you for this interesting question. We believe that a prehabilitation program of at least three weeks prior to primary cytoreductive surgery would only minimally reduce eligbility rates. Primary cytoreductive surgery should ideally take place as fast as possible after the ovarian cancer diagnosis. However, it normally takes a few weeks before ovarian cancer patients undergo primary cytoreductive surgery after their diagnosis due to adjuvant radiologic imaging, multidisciplinary consultations and preoperative assessment. This should enable undergoing a prehabilitation program before surgery with at most a minimal delay, if any, without significantly affecting eligbility rates. Besides that, most patients underwent interval cytoreductive surgery and not primary surgery. As interval cytoreductive surgery is preceded by neoadjuvant chemotherapy, this definitely gives a clear window of opportunity for a prehabilitation program. We have added an extra paragraph in our discussion (page 11, paragraph 3, line 367-73).
Comments 2: Could the Authors report the possible reasons for better tolerance to chemotherapy in terms of dose reductions and dose deferrals between prehabilitation patients and control groups?
Response 2: In literature, the relationship between prehabilitation and better chemotherapy tolerance is described in patients with esophagogastric and breast cancer, but not yet in ovarian cancer. However, it is known that prehabiltation leads to improved cardiopulmonary fitness and therefore better resilience as shown in our article and in literature. This could explain the improved chemotherapy tolerance. Therefore, we have added an extra sentence in the discussion (page 11, paragraph 5, line 389-91).
Reviewer 2 Report
Comments and Suggestions for Authors
The manuscript addresses an important clinical topic—multimodal prehabilitation for ovarian cancer patients undergoing cytoreductive surgery. The study is well-structured, with a clear research question, appropriate methodology, and clinically relevant findings. The results suggest that prehabilitation is both feasible and effective in improving functional capacity and chemotherapy tolerance. However, there are several areas where the manuscript could be strengthened, including methodological clarifications, statistical analyses, and discussion depth.
- The study has a relatively small sample size, which may limit the statistical power to detect differences in postoperative outcomes.
- The authors should consider conducting a power analysis to justify whether the study was adequately powered to detect meaningful clinical effects.
- If feasible, additional recruitment or pooling data from similar studies could strengthen conclusions.
- The study reports no significant differences in postoperative complications, readmissions, or ICU stays between prehabilitation and control groups.
- The discussion should address why the expected benefits of prehabilitation (e.g., reduced complications) were not observed.
- Were there confounding variables such as differences in surgical complexity, preoperative health status, or perioperative care?
- The study does not report on long-term outcomes such as progression-free survival or quality of life.
- If follow-up data are available, including these outcomes would enhance the clinical relevance of the findings.
- If not available, the limitations should be explicitly stated.
- Some tables present a large volume of numerical data, making them difficult to interpret quickly.
- Highlighting key findings in bold or adding summary annotations would improve clarity.
- The manuscript mentions ongoing trials (SOPHIE, PROPER, KORE-INNOVATION).
- Specifying how this study complements or differs from these trials would be valuable.
Author Response
Thank you very much for the nice words on the relevance of our manuscript and for taking the time to review it. Please find our detailed responses below and the corresponding revisions, marked in red, in the re-submitted files. We expect that your comments have improved the manuscript sufficiently to merit consideration for publication.
Comments 1: The study has a relatively small sample size, which may limit the statistical power to detect differences in postoperative outcomes. The authors should consider conducting a power analysis to justify whether the study was adequately powered to detect meaningful clinical effects.
Response 1: Thank you for pointing this out. As the primary goal of this study was to investigate the feaasibility of implementing prehabilitation, we decided to not perform an a priori power analysis. However, we do agree with the fact that a power analysis justifies whether the study was adequately powered to detect meaningful clinical effects. Therefore, we have added a post-hoc power analysis of the postoperative outcomes in the results section (page 10, paragraph 1, line 329-31), and a description of this analysis in the methods section (page 6, paragraph 6, line 216-9). We also adjusted the corresponding paragraph in the discussion (page 11, paragraph 4, line 374-9).
Comments 2: If feasible, additional recruitment or pooling data from similar studies could strengthen conclusions.
Response 2: We appreciate this suggestion and agree that this could strengthen our manuscript. Unfortunately, both options were not considered feasible for this manuscript. Given the upcoming interest in prehabilitation, we believe our results are significant enough to publish at this stage because of the uniqueness of the multicenter aspect and outcomes on adjuvant chemotherapy tolerance. We did not consider pooling data from similar studies appropriate, as our goal was to give a detailed report about the real-life implementation of prehabilitation in two top clinical oncology hospitals, and are the first to report on a multicenter setting. Adding data from other studies would not match this objective. Besides that, each prehabiltation program is slightly differently and each hospital has their own clinical practice, making it hard to compare these data.
Comments 3: The study reports no significant differences in postoperative complications, readmissions, or ICU stays between prehabilitation and control groups. The discussion should address why the expected benefits of prehabilitation (e.g., reduced complications) were not observed.
Response 3: Thank you for pointing this out. As we already stated in our article, we think this is the result of the small sample size and reduced statistical power consequently, rather than a real absence of effects. We clarified this point in our discussion (page 11, paragraph 4 , line 374-9) and have added the aforementoined post-hoc power analysis to emphasize this.
Comments 4: Were there confounding variables such as differences in surgical complexity, preoperative health status, or perioperative care?
Response 4: As shown in Table S1 in the Supplementary materials, there were no statistical differences in surgical complexity or preoperative health status between patient groups within the same hospital. There were only slight differences in perioperative care between the two hospitals. For instance, patients at the Albert Schweitzer Hospitals are generally discharged earlier after cytoreductive surgery. However, as the emphasis of this article is about the differences between prehabiltation who did and did not receive prehabilitation rather than on differences between the two hospitals, we chose not to elaborate further on these differences, as they are no confounding variables.
Comments 5: The study does not report on long-term outcomes such as progression-free survival or quality of life. If follow-up data are available, including these outcomes would enhance the clinical relevance of the findings. If not available, the limitations should be explicitly stated.
Response 5: Thank you for this interesting suggestion. We agree that adding long-term outcomes like progression-free survival would enhance the clinical relevance of the manuscript. However, these data would require a follow-up of several years and are unlikely to be statistically meaningful, given our small sample sizes. Therefore, we have decided to focus primarily on the implementation and feasibility of the prehabilitation program. We did consider to include patients’ quality of life by organizing focus groups to explore their experiences with the process. However, this would require a seperate qualitative methodological approach. As our manuscript already combines two methodologies (reporting feasibility and qualitative assessments of effects and differences), we believe it would be more appropriate to address the qualitative aspects in a future, dedicated publication. We have added two extra sentences to the study limitations in our discussion (page 12, paragraph 1, line 407-9).
Comments 6: Some tables present a large volume of numerical data, making them difficult to interpret quickly. Highlighting key findings in bold or adding summary annotations would improve clarity.
Response 6: We agree that the tables contain a lot of data, as we had a lot of data to present. We have made some changes to clarify the tables. In Table 1 and 2, we have added extra vertical lines and an asterisk at all signficant findings, which were already in bold (Table 1: page 9, paragraph 3, line 309; Table 2: page 10, paragraph 3, line 335-6). In table S1 in the Supplementary materials, we have added extra vertical and horizontal lines.
Comments 7: The manuscript mentions ongoing trials (SOPHIE, PROPER, KORE-INNOVATION).
Specifying how this study complements or differs from these trials would be valuable.
Response 7: Thank you for pointing this out. These trials are all randomized and include a large sample size (≥ 140 patients). Given their design and sample size, these trials are expected to be of high methodological quality and are expected to adequately detect differences in postoperative outcomes with enough statistical power, unlike our study. However, our study is unique in describing the feasibility and real-time implementation of prehabilitation, and in presenting results about adjuvant chemotherapy tolerance. We have added two extra sentences in this paragraph (page 12, paragraph 3, line 420-4).
Reviewer 3 Report
Comments and Suggestions for Authors
This study assessed the viability and outcomes of two distinct multimodal prehabilitation programs that were put in place prior to cytoreductive surgery (CRS) in two Dutch hospitals. However, there are certain concerns need to be addressed before considering it for publication.
- The abstract mentions "multimodal prehabilitation," but it is unclear what specific components were included (e.g., physical, nutritional, psychological). A detailed breakdown is essential.
- Given that the two hospitals' initiatives differed in "timing, extent, and content," how was it justifiable to compare data from these disparate interventions?
- With only 32 patients in the intervention group, was a power analysis conducted beforehand? Is the sample size sufficient to detect significant differences in postoperative outcomes?
- Since non-adherent patients are not included, per-protocol analysis may create bias even though it is a viable method for evaluating efficacy. For transparency, the authors are able to provide both intention-to-treat (ITT) and per-protocol analyses.
- The handling of missing data was not mentioned. The authors can indicate whether any exclusion criteria or imputation techniques (mean substitution, multiple imputation) were used.
- Did the authors test for normality before using the unpaired or paired t-tests?
- In the two hospitals, only 6 out of 11 and 8 out of 11 patients had complete paired data, which introduced bias and decreased statistical power. It is advised that the authors explain why some endpoint measures were absent and address the implications of missing data.
- The 6MWT distance did not achieve relevance, despite a notable improvement in upper body strength. This might indicate a shorter prehabilitation period or a less rapid conversion of strength to endurance. explain this disparity between gains in functional mobility and strength.
- Are there any plans to increase standardization between sites in future studies to reduce inter-site variability?
- How do the authors propose integrating patient and healthcare professional feedback into program design? Will this involve structured interviews, validated questionnaires, or implementation tools like RE-AIM or CFIR frameworks?
Author Response
Thank you very much for taking the time to review this manuscript. Please find the detailed responses below and the corresponding revisions, marked in red, in the re-submitted files. We hope these revisions have strengthened the manuscript sufficiently to merit consideration for publication.
Comments 1: The abstract mentions "multimodal prehabilitation," but it is unclear what specific components were included (e.g., physical, nutritional, psychological). A detailed breakdown is essential.
Response 1: Thank you for this comment. We have added the following sentence in the abstract (page 1, paragraph 2, line 28-9): “The programs included at least physiotherapy, dietary advice and intoxication cessation.”
Comment 2: Given that the two hospitals' initiatives differed in "timing, extent, and content," how was it justifiable to compare data from these disparate interventions?
Response 2: The prehabiltiation programs differed in timing, extent and content, making direct comparison between the two hospitals difficult. Therefore, we did not compare outcomes between hospitals, but only within each hospital. We only combined data from both hospitals for patients who had both baseline and endpoint measurements of the physiotherapy trajectory available, solely to illustrate that the lack of statistically significant results in each hospital individually was likely due to the small sample sizes, as the percentage changes were comparable.
Comments 3: With only 32 patients in the intervention group, was a power analysis conducted beforehand? Is the sample size sufficient to detect significant differences in postoperative outcomes?
Response 3: No sample size calculation or a priori power analysis was made as the primary objective of this study was to evaluate feasibility. However, we decided to add a post-hoc power analysis to investigate whether this study was adequately powered to detect significant changes in postoperative outcomes. The outcomes of the post-hoc power analysis are in the results section (page 10, paragraph 1, line 329-31), and a description of this analysis in the methods section (page 6, paragraph 6, line 216-9). We also adjusted the corresponding paragraph in the discussion (page 11, paragraph 4, line 374-9).
Comments 4: Since non-adherent patients are not included, per-protocol analysis may create bias even though it is a viable method for evaluating efficacy. For transparency, the authors are able to provide both intention-to-treat (ITT) and per-protocol analyses.
Response 4: Thank you for pointing this out. We acknowledge that a per-protocol analysis may introduce bias. However, we did not consider an intention-to-treat (ITT) analysis suitable for this manuscript, given that the primary objective of this study was to assess the feasibility of implementing a prehabilitation program. When evaluating feasibility, only a per-protocol analysis was suitable, as an ITT would not have accurately reflected the real-life implementation of the program, as it would include patients who did not actually participate in the program.
Comments 5: The handling of missing data was not mentioned. The authors can indicate whether any exclusion criteria or imputation techniques (mean substitution, multiple imputation) were used.
Response 5: Thank you for pointing this out. We excluded missing data and did not use imputation techniques. We had two types of missing data in our manuscript. At first, we missed some baseline characteristics. Given the limited extent and non-systematic nature of this missing data, we chose not to apply imputation techniques. Since the missing baseline characteristics were not related to the primary outcomes and affected only a few cases, we considered the potential impact on our results to be minimal. Secondly, we missed data because some patients did not undergo physiotherapy treatments, surgery or adjuvant chemotherapy. As this was mostly the result of clinical decisions and therefore missing not at random, we did not apply any data imputation techniques. We have now added an extra sentence in the Methods section (page 6, paragraph 6, line 219).
Comments 6: Did the authors test for normality before using the unpaired or paired t-tests?
Response 6: We tested for normality before using the t-tests, and added this in the methods. (page 6, paragraph 6, line 209-10 and 214).
Comments 7: In the two hospitals, only 6 out of 11 and 8 out of 11 patients had complete paired data, which introduced bias and decreased statistical power. It is advised that the authors explain why some endpoint measures were absent and address the implications of missing data.
Response 7: Thank you for pointing this out. We have added why some endpoint measurements were missing in the results (page 8, paragraph 2, line 288-90). Those data are missing not at random data, which can lead to bias and decreased statistical power. We have added this in the discussion (page 12, paragraph 1, line 405-9).
Comments 8: The 6MWT distance did not achieve relevance, despite a notable improvement in upper body strength. This might indicate a shorter prehabilitation period or a less rapid conversion of strength to endurance. explain this disparity between gains in functional mobility and strength.
Response 8: Thank you for this interesting question. We think that the non-significant change in 6MWT distance is a result of the small sample size, rather than a true absence of effect. As the prehabilitation programs had a revalitely short duration, it could be plausible that only improvements in muscle strength would be seen and not in endurance, as endurance usually requires a longer timeframe to improve. However, we did observe significant changes in the MSEC on the Steep Ramp Test, which also reflects endurance, suggesting that improvements in endurance were made despite the limited timeframe.
Comments 9: Are there any plans to increase standardization between sites in future studies to reduce inter-site variability?
Response 9: This is a good suggestion for the discussion of our manuscript. In the Netherlands, there already is a detailed protocol about the desired implementation of a prehabilitation program before surgery for colorectal cancer. It would be a good idea to develop a standardized protocol for ovarian cancer patients as well to reduce inter-site variability and ensure consistent quality of care. We have added this recommendation in our discussion (page 12, paragraph 4, line 428-31).
Comments 10: How do the authors propose integrating patient and healthcare professional feedback into program design? Will this involve structured interviews, validated questionnaires, or implementation tools like RE-AIM or CFIR frameworks.
Response 10: Thank you for this interesting question. We think that semi-structured interviews conducted in focus groups are the most suitable way to gather valuable feedback from patients and health care professionals regarding the implementation of prehabilitation. This gives an opportunity for in-depth questioning, while still providing enough structure to ensure that all relevant topics are addressed. We think that implementation tools like RE-AIM of CFIR are valuable to implement the changes that will be deemed necessary based on the semi-structured interviews. We have added these suggestions in the discussion (page 12, paragraph 4, line 426-7).
Round 2
Reviewer 3 Report
Comments and Suggestions for Authors
The quality of the paper is improved